# A Genome-Focused Investigation Reveals the Emergence of a *Mycobacterium tuberculosis* Strain Related to Multidrug-Resistant Tuberculosis in the Amazon Region of Brazil

**DOI:** 10.3390/microorganisms12091817

**Published:** 2024-09-02

**Authors:** Emilyn Costa Conceição, Johannes Loubser, Arthur Emil dos Santos Guimarães, Abhinav Sharma, Liliana Kokusanilwa Rutaihwa, Anzaan Dippenaar, Richard Steiner Salvato, Ricardo José de Paula Souza e Guimarães, Maria Cristina da Silva Lourenço, Wandyra Araújo Barros, Ninarosa Calzavara Cardoso, Robin Mark Warren, Sebastien Gagneux, Beatriz Gilda Jegerhorn Grinsztejn, Philip Noel Suffys, Karla Valéria Batista Lima

**Affiliations:** 1Programa de Pós-Graduacao em Pesquisa Clinica e Doencas Infecciosas, Instituto Nacional de Infectologia Evandro Chagas, Fundacao Oswaldo Cruz, Manguinhos, Rio de Janeiro 21046-360, RJ, Brazil; gbeatriz@ini.fiocruz.br; 2Department of Science and Innovation-National Research Foundation Centre of Excellence for Biomedical Tuberculosis Research, South African Medical Research Council Centre for Tuberculosis Research, Division of Molecular Biology and Human Genetics, Faculty of Medicine and Health Sciences, Stellenbosch University, Cape Town 7505, Western Cape, South Africa; jloubser@sun.ac.za (J.L.); abhinavsharma@sun.ac.za (A.S.); anzaan.dippenaar@uantwerpen.be (A.D.); cristina.lourenco@ini.fiocruz.br (M.C.d.S.L.);; 3Programa de Pos-Graduacao Biologia Parasitaria na Amazonia, Universidade do Estado do Para, Belém 66075-110, PA, Brazil; arthuremil@hotmail.com; 4Swiss Tropical and Public Health Institute, 4123 Allschwil, Switzerland; liliana.rutaihwa@finddx.org (L.K.R.); sebastien.gagneux@unibas.ch (S.G.); 5University of Basel, 4001 Basel, Switzerland; 6Family Medicine and Population Health, Faculty of Medicine and Health Sciences, University of Antwerp, 2610 Antwerp, Belgium; 7Programa de Pós-Graduacao em Biologia Celular e Molecular, Universidade Federal do Rio Grande do Sul, Porto Alegre 90010-150, RS, Brazil; richardsalvato@hotmail.com; 8Secao de Epidemiologia, Instituto Evandro Chagas, Ananindeua 67030-000, PA, Brazil; ricardojpsg@gmail.com; 9Laboratório de Bacteriologia e Bioensaios em Micobacterias, Instituto Nacional de Infectologia Evandro Chagas, Fundacao Oswaldo Cruz, Manguinhos, Rio de Janeiro 21046-360, RJ, Brazil; 10Hospital Universitario Joao de Barros Barreto, Universidade Federal do Pará, Belém 66073-000, PA, Brazil; wandy@ufpa.br; 11Laboratório de Biologia Molecular Aplicada a Micobacteria, Instituto Oswaldo Cruz, Fundacao Oswaldo Cruz, Rio de Janeiro 21046-360, RJ, Brazilpsuffys@gmail.com (P.N.S.); 12Seção de Bacteriologia e Micologia, Instituto Evandro Chagas, Ananindeua 67030-000, PA, Brazil

**Keywords:** tuberculosis, multidrug-resistant, whole-genome sequencing, genotype emergence, MIRU-VNTR, lineage

## Abstract

A previous study in Pará, Northern Brazil, described a strain of *Mycobacterium tuberculosis* with a unique genotype (SIT2517/T1) associated with multidrug-resistant tuberculosis (MDR-TB). To improve our understanding of MDR-TB transmission dynamics of these strains within this region, we performed phenotypic and genotypic drug susceptibility testing (pDST/gDST), 24-loci mycobacterial interspersed repetitive units (MIRU-VNTR) genotyping, whole-genome sequencing (WGS) and geo-epidemiology analysis. Of the 28 SIT2517/T1 isolates, 19 (67.9%) could be genotyped by 24-loci MIRU-VNTR and 15 by WGS. All belonged to sublineage 4.1.1.3, distinct from other representative Lineage 4 isolates identified in Brazil. The MDR phenotype determined by pDST was confirmed by gDST, the latter also demonstrating the presence of additional mutations conferring pre-extensively drug-resistance (pre-XDR). Discrepancies between gDST and pDST were observed for pyrazinamide and fluoroquinolones. Thirteen out of 15 isolates analyzed by WGS were clustered when applying a 12 single nucleotide polymorphisms (SNPs) cutoff. The SIT2517/T1 isolates were distributed across the metropolitan regions of Belém and Collares municipalities, showing no geographic clustering. WGS-transmission network analysis revealed a high likelihood of direct transmission and the formation of two closely linked transmission chains. This study highlights the need to implement TB genomic surveillance in the Brazilian Amazon region.

## 1. Introduction

Although genomic-epidemiology surveillance has been proposed as a valuable strategy for monitoring and controlling infectious diseases [1,2], its implementation remains limited, primarily due to economic constraints [3]. If embraced on a larger scale, health agencies responsible for screening and interventions could gather the essential information needed to develop methodologies to prevent and treat diseases with pandemic or epidemic potential. Moreover, it would enable the identification of areas for which health policies can be enhanced and optimized [1,2].

One such disease is tuberculosis (TB), a preventable and curable disease that remains one of the world’s deadliest infectious killers and is defined as a global epidemic [4]. In efforts to understand and describe the evolution of *Mycobacterium tuberculosis* (*Mtb*) and unravel transmission networks, genotyping methods like mycobacterial interspersed repetitive units (MIRU-VNTR), restriction fragment length polymorphism (RFLP), or spoligotyping have been used in the past [5,6,7]. Each of these methods identifies certain genomic patterns unique to strains but remains limited by only observing a fraction of the genome. This is more circumvented by the use of whole genome sequence (WGS) but despite the release of the first genome of *Mtb* already been established in 1998 [8], lower-income countries often still rely on the use of these older genotyping methods since it is inexpensive and able to accurately distinguish between distantly related strains.

The use of these methods, however, sometimes results in incorrect clustering of closely related strains. In these cases, WGS can be used for a more accurate estimation of genetic distance due to its high resolution. Further benefits of WGS are the accurate prediction of lineages, sub-lineages, and drug resistance data that could be used in the future to understand virulence, pathogenicity and resistance mechanisms. All of this information, in conjunction with the current gold standard for drug susceptibility testing (DST) [9] provides more comprehensive data that can be used by clinicians to administer more accurate treatment, especially in high-burden settings. WGS allows for accurate clustering and transmission dynamics investigation, improved diagnosis, surveillance and source investigation [10].

The use of WGS in *Mtb* surveillance and research has demonstrated that the transmission of drug-resistant TB (DR-TB), rather than poor or insufficient treatment leading to the acquisition of drug resistance, is one of the major driving forces of the DR-TB epidemic [11,12,13]. Transmission of DR-TB is a direct threat to public health and should be prioritized [14]. DR-TB can be classified into five categories according to the World Health Organization (WHO): (i) isoniazid-resistant TB (HR-TB), (ii) rifampicin-resistant TB (RR-TB), (iii) multidrug-resistant TB (MDR-TB) (simultaneously resistant to isoniazid and rifampicin–the two most effective first-line anti-TB drugs), (iv) plus pre-extensively drug-resistant TB (pre-XDR-TB), which is resistant to rifampicin and any fluoroquinolone, a class of second-line anti-TB drug) and (v) extensively drug-resistant TB (XDR-TB), which is simultaneously resistant to rifampicin, any fluoroquinolone and either bedaquiline or linezolid [15,16].

In previous genotyping studies conducted in Brazil, we described region specific presence of *Mtb* isolates of the East African Indian (EAI) genotype in Pará, Northern Brazil [17]. Another particular genotype was observed in 28 isolates in this region only is the Shared International Type (SIT) 2517 sublineage T1 (SIT2517/T1), having so far been described exclusively in Brazil [18]. Based on phenotypic DST (pDST), all but one of these isolates were MDR.

To gain further insight into the transmission dynamics of this specific genetic group, we performed additional genotyping by using MIRU-VNTR and WGS which provide increased resolution compared to Spoligotyping [17]. Additionally, the application of WGS to detect and study the transmission of DR-TB isolates is gaining increasing importance within TB surveillance programs. To date, only a few studies in Brazil have implemented WGS to study the epidemiology and genomic diversity of *Mtb* [19,20]. The largest and most recent study comprised *Mtb* strains isolated from 2014 to 2019 highlighting the contribution of prisons in disseminating TB in Brazil [21] and the need for increased WGS surveillance.

Due to the high cost of WGS, low-income countries often rely on genotyping tools like MIRU-VNTR for tracking the spread of diseases [5,22,23]. While WGS is considered more powerful, and capable of resolving clusters than 24-MIRU-VNTR [6,7], this advantage mostly applies to highly similar isolates [24,25]. We therefore used both techniques to evaluate the extent of transmission and distribution of the emerging *Mtb* genotype (SIT2517/T1) associated with MDR-TB in Pará/Brazil

## 2. Materials and Methods

### 2.1. Ethical Statement

This study was approved by the Ethical Committee of the Evandro Chagas Institute IEC) Ananindeua-Pará, number 059750/2017, CAAE: 69248217.0.0000.0019.

### 2.2. Study Population and Epidemiological Information

Between 1998 and 2010, we conducted a large population-based study on 980 *Mtb* isolates isolated obtained from pulmonary TB patients in Pará, Brazil. All isolates underwent spoligotyping and 28 isolates exhibiting SIT 2517/T1 profile were selected for the present analysis [17]. Clinical and epidemiological information, including demographic data, was retrieved from patient records. Information on TB contact was collected through routine interviews conducted for TB monitoring at the reference University Hospital João de Barros Barreto (HUJBB).

### 2.3. Culture, Drug Susceptibility and Molecular Tests

Bacterial culture, pDST, DNA isolation, and manual spoligotyping were performed during a previous study by Conceição et al. [17]. We performed semi-automated 24-loci MIRU-VNTR typing using a QIAGEN kit (HotStarTaq DNA polymerase, QIAGEN, Hilden, Germany). After the PCR reaction, amplicons were analyzed using the ABI3130 instrument (Applied Biosystems, Waltham, USA) at the Evandro Chagas Institute in Pará [26]. The resulting alleles were assigned numerical values according to the number of repeats in the GeneMapper software v3.7 (Applied Biosystems, Waltham, MA, USA). We used the MIRU-VNTRplus database (https://www.miru-vntrplus.org/MIRU/index.faces, accessed on 27 August 2024) for lineage identification based on genotype positioning within a Neighbor-Joining [27].

### 2.4. Whole-Genome Sequencing

The genomic DNA was quantified using a Qubit double-strand DNA (dsDNA) broad range (BR) assay kit Thermo Fisher Scientific, Waltham, MA, USA). Subsequently, the DNA was diluted to a concentration ranging between 0.8 to 1 ng/uL. Libraries were prepared using the Nextera XT DNA Library Preparation Kit (Illumina, San Diego, CA, USA) according to the manufacturer’s instructions. The sequencing was conducted on an Illumina HiSeq 2500 instrument (Illumina, San Diego, CA, USA) at the Swiss Tropical and Public Health Institute in Basel, Switzerland.

### 2.5. Bioinformatic Analysis of Genomic Data

The WGS analysis was performed using an in-house and automated pipeline Universal Sequence Analysis Pipeline (USAP). This pipeline encompassed sequence quality control, alignment to the reference genome, variant identification, and genome annotation.

Briefly, Trimmomatic v. 0.32 [28] was used to trim the raw reads based on quality. High-quality bases and reads were selected as input for alignment to the reference genome *Mtb* H37Rv (GenBank NC000962.3) using Burrows-Wheeler Aligner (BWA) v0.6.2 [29], Novoalign (Novocraft) v3.02.13 and SMALT tools v0.7.5. Subsequently, single nucleotide variations (SNVs), including single nucleotide polymorphisms (SNPs) and small insertions and deletions (indels), were detected using GATK v3.5 and SAMTools v1.3. Variants identified by both variant callers in all three alignments were annotated and utilized in subsequent analysis. The genome coverage and large deletions were assessed using the BEDTools software package v. 2.2.25 [30], which identifies regions with zero depth of coverage, indicating potential genomic deletions referred to as regions of difference (RDs). SNVs between isolates were determined by manual pairwise comparisons.

For phylogenetic analysis, the variant files were filtered to exclude variants identified in hard-to-map and repetitive regions such as PE/PPE, insertion and phage sequences while maintaining only variable sites identified at a frequency cutoff of 90% and minimum depth of 30X. These high-confidence variable sites were (*n* = 32,927) used to create a multi-FASTA file that was used to construct a maximum likelihood phylogenetic tree using IQ-TREE 2.0.6 model [31]. We used the Interactive Tree of Life (ITOL), https://itol.embl.de/ (accessed on 27 August 2024) to visualize the phylogenetic tree [32]. To enhance the analysis, we included *Mtb* Lineage 4 genomes from other studies conducted in Brazil [19,33,34,35,36,37]. Appendix A provides the accession identifications of these included samples.

For the in-silico prediction of anti-TB drug susceptibility profiles, also known as genotypic DST (gDST), we employed TBProfiler v6.2.1 [38]. This classification of the drug resistance profile was based on new definitions provided by the WHO Catalogue of Mutations 2nd version [15,16,39]. The high-confidence drug resistance markers identified for each isolate are presented in Appendix A.

### 2.6. Geographic Information System

Geographic Information System (GIS) mapping was performed from the place of residence for each patient and plotted as geographical coordinates (latitude and longitude) using Google Earth software v.7.3 (https://www.google.com/earth/, accessed on 20 April 2024) and a hand-held Global Positioning System receiver. The geoprocessing was done at the Geoprocessing Laboratory, IEC.

### 2.7. Transmission Analysis

Transmission analysis was performed by TransFlow (v1.0) [40] by using the date of collection of the isolate, sex, age and residential location (longitude and latitude) of each patient. Briefly, sequences were trimmed by Trimmomatic (v0.36), quality control checked by FastQC (v0.119) and aligned to a computational pan-genome consisting of 146 lineages including Lineages 1 to 4 *Mtb* genomes by BWA (v0.7.17) and variants called by GATK (v3.8) [41]. Pairwise SNP-distance calculations, transmission clustering and transmission network building were performed with PANPASCO, TransCluster (v0.1.0) and the SeqTrack (v1.3) algorithm from Adegenet (v2.1.7) in R (v4.0.5). Variants identified in hard-to-map and repetitive regions as well as those associated with drug resistance were omitted before constructing the transmission network. The network transmission visualizations were generated by the ggnet2 function from GGally (v2.1.2).

## 3. Results

Among 28 isolates of *Mtb* with genotype SIT2517/T1, 19 isolates (67.9%) were retrieved for further genotyping using a 24-loci MIRU-VNTR. Out of 19 isolates, 10 possessed a complete set of 24 loci. The remaining isolates had varying degrees of incompleteness: six lacked two loci, two lacked one, and one isolate was missing a significant portion. The remaining nine isolates (32.2%) could not be genotyped due to either low DNA integrity (*n* = 5; 17.9%) or the unavailability of DNA samples (*n* = 4; 14.3%). We attempted subculture for these nine isolates but were unsuccessful. Among the 19 isolates, 17 underwent WGS, and two failed to pass quality control (coverage <20X) and were excluded from further WGS analysis. Therefore, a complete bioinformatic analysis was conducted on 15 isolates (54%) (Table 1).

We associated these isolates with socio-demographic and epidemiological data and constructed the patient epidemiological link based on three main scenarios: classical epidemiology (Figure 1A); molecular approach based on 24-loci MIRU-VNTR (Figure 1B) and based on WGS analysis (Figure 1C).

The phylogenetic tree, based on high-confidence variable sites identified through WGS of 15 *Mtb* genomes, including at least three representatives from all lineages and ancestral strains as well as 72 *Mtb* Lineage 4 sequences from Brazil (Appendix A), demonstrates that the SIT2517/T1 isolates form a separate cluster (single clade) (Appendix A) and belong to the sublineage 4.1.1.3. According to the gDST, all isolates were at least MDR-TB, carrying the mutations rpoB_*p*.Ser450Leu and katG_*p*.Ser315Thr. Isolate G21387 had an additional mutation c.-8T>C upstream of the *fabG1* gene that has been associated with isoniazid and ethionamide resistance [15,16], [39].

All isolates harboured the mutation pncA_c.464dupT, known to cause pyrazinamide resistance. Isolate G21390 was classified as pre-XDR because of the additional mutations embA_c.-11C>A and embB_*p*.Gly406Ser, associated with ethambutol-resistance, gyrA_*p*.Asp94Asn (fluoroquinolone-resistance) and ethA_c.935dupT (ethionamide- resistance). All isolates also harboured the three mutations *gyrA* (gyrA_*p*.Glu21Gln, gyrA_*p*.Ser95Thr, gyrA_*p*.Gly668Asp) that are not directly linked to fluoroquino resistance according to the updated WHO catalogue [39]. Nineteen additional variants in or upstream of other drug-resistance-associated genes but not confirmed to be associated with drug resistance were detected in all isolates (Appendix A). These fixed variants were detected in genes associated with resistance to rifampicin (rpoB_*p*.Ala599Val, rpoC_*p*.Gly594Glu, Rv2752c_*p*.Gly137Cys), ISONIAZID (Rv2752c_*p*.Gly137Cys), ethambutol (embA_c.-590C>T, embC_*p*.Val981Leu, embB_c.2895G>A), streptomycin (rpsL_c.-165T>C), kanamycin/amikacin/capreomycin (rrs_*n*.-187C>T, whiB6_c.75delG, whiB6_c.-211C>T), ethionamide (Rv0565c_*p*.Ser68Pro, Rv3083_c.574T>C), cycloserine (ald_c.-32T>C), linezolid (rplC_c.-590T>C) and bedaquiline and clofazimine (mmpL5_*p*.Ile948Val, Rv1979c_c.-129A>G). Further fixed variants also classified as ‘’other’’ but not detected in all the isolates were whiB6_c.-124G>T (G21383, G21376, G21384, G21388, G21390, G21391, G21393) and Rv3083_*p*.Trp261* (G21390) (Appendix A). 

Based on gDST, pyrazinamide resistance was detected in all isolates, while pDST did not detect this resistance in seven isolates (G21376, G21386, G21388, G21389, G21390, G21391 and G21393). As pDST to fluoroquinolones was not performed, it was not possible to compare gDST/pDST results.

Analyzing the geospatial distribution determined by the patients’ residences, we noted their dispersion throughout the Belem and Colares municipalities. When examining the yearly geographical distribution, as well as the DST, MIRU-VNTR, and WGS profiles, there is no specific geographical hotspot. Nevertheless, samples with similar genetic profiles tended to be more closely distributed. Noteworthy is that the drug-susceptible isolate (2522) was isolated from a patient resident of the municipality of Colares and according to MIRU-VNTR typing (no WGS available) belonged to Cluster 3 including mostly patients’ residents of the metropolitan region of Belem (Figure 2).

The transmission network structure indicates G21384 as the most likely isolated from the index case of at least four transmission events: towards G21376 (4 SNPs), G21388 (0 SNPs), G21390 (11 SNPs) and G21379 (6 SNPs). The number indicated in brackets refers to the number of variants identified by the in-house pipeline through manual pairwise comparisons (Appendix A). G21384, isolated in 2000, was obtained from the Godfather of another patient (pairwise isolate 1694, not sequenced) with 0 SNPs distance from G21388, isolated in 2007, with unknown TB contact.

Isolate G21379 was linked to at least five transmission events, one being a household contact (G21377, sister-0 SNPs), which in turn is linked to another household member (G21383, sister-0 SNPs). Within this study population, we observed that the TB transmission was multidirectional from the individual infected by isolate G213179: from G213179 to G21389 (isolated from a father of a different household-1 SNPs) and another direct transmission then occurs (G21392, son-2 SNPs), and from G21379 to G21382, G21386 and G21385 (unknown contact, 0 SNPs).

The patient infected by isolate G21390 appears to have transmitted at least once (to G21391, unknown contact-6 SNPs) prior to evolving towards pre-XDR according to gDST. From the patient infected by G21391, there was a direct transmission to G21393 (unknown contact-0 SNPs). Only one isolate, G21387, seems distantly linked to the rest of the isolates of this bi-part transmission network, with a SNP distance of 12 between this isolate and G21385 and distances greater than 12 compared to other isolates. This was the only isolate with an additional mutation in the *fabG1* gene associated with isoniazid and ethionamide resistance.

## 4. Discussion

The WGS approach is crucial for pathogen surveillance [2], for offering rapid identification of TB-DR and to treat patients more accurately with shorter, more effective and less toxic anti-TB regimens [42]. Although in general, outbreak studies of *Mtb* by WGS have been retrospective [43], it has been pivotal for a better understanding of transmission dynamics, evolution of the pathogen and identification of resistance markers [10]. One major finding driven by WGS analysis has been that the transmission of MDR plays a major role in the global epidemic, rather than poor treatment [13]. 

We therefore further characterized using MIRU-VNTR and WGS analysis, a specific set of SIT2517/T1 isolates that has so far been described only in the state of Pará, Brazil and associated with MDR-TB, (Appendix A). Except for two, all isolates formed part of a transmission chain (<12 SNPs), with some most likely indicative of recent direct transmission (Figure 1C, Appendix A).

Our study analyzed 28 *Mtb* SIT2517/T1 isolates obtained from a retrospective analysis of 980 archived MTBC isolates at the IEC from 1998 to 2011. This represents a subset of the total TB cases in Pará state during this period, as the IEC processed 100% of cultures for pDST for the state with an average TB incidence of 39 cases per 100,000 inhabitants [17]. Moreover, the 980 patients referred for culture and drug susceptibility testing at the IEC likely represented suspected cases of TB-DR or contacts of known TB patients. Consequently, the study design focused on this specific population and may have missed a broader picture of SIT2517 prevalence within the general TB population of Pará. This limitation highlights the importance of performing comprehensive genomic surveillance for TB, especially with the emergence of new treatment options. All the isolates were classified as MDR, except for one that has evolved further to be pre-XDR. This is noteworthy, as the transmission of MDR-TB is one of the major drivers of outbreaks and epidemics in many countries. Additionally, after about 15 years, these SIT2517 MDR strains may have acquired resistance to other antibiotics, potentially evolving into pre-XDR or XDR.

Based on SNPs distance approach for transmission network analysis, strain G21384 (isolated in 2000) appeared to have originated from the index case of at least four transmission events. This strain showed 0 SNP difference from G21388 (isolated in 2007) with unknown epidemiological linkage. However, it is known that G21384 is related to NP1694 (not sequenced), differing in the MIRU-VNTR result by a single locus out of 24 loci. The lack of SNPs between strains isolated seven years apart could be explained from different perspectives. Besides the fact that TB cases might be missing in this network, within-host microevolution of *Mtb* can lead to subpopulations that differ from each other by more than 12 SNPs, since the genomic variability within a TB patient may exceed that observed between any two epidemiologically linked cases [44].

In addition to analyzing the consensus SNPs, the examination of heterogeneous alleles (hSNPs) has been proposed to improve the resolution of transmission inference. This approach is particularly relevant due to within-host Mtb diversity, which allows different scenarios such as the transmission of minority alleles from an index case to a secondary case that does not necessarily impact the consensus SNPs. Nevertheless, further evidence is required regarding the role of the transmission associating hSNPs with confident sequencing depth (deep sequencing) [45,46]. Furthermore, applying a ≤12 SNP cutoff for defining transmission chains is contingent upon the local epidemiological landscape. For instance, the interpretation of this threshold may vary. Supporting this variability, a separate study investigating household contacts infected with *Mtb* Lineage 1 within the same region (State of Pará) observed robust epidemiological links even among cases exceeding the 12 SNP threshold [47]. Additionally, the presence of identical drug-resistance-associated variants shared amongst isolates that are within these clusters further strengthens the clustering using the presently used cutoff [48].

It is important to highlight that the particular epidemiological links were observed in this study cohort because they were referred to the HUJBB which is the reference for TB-DR and contact tracing, also present in the previously mentioned study [17] that identified by spoligotyping the presence of Lineage 1 (East-African Indian) uncommonly present in South America but described in about 10% of the TB cases from 1998 to 2011, also referred to the HUJBB.

In the Amazonian area, there are limited studies about *Mtb* genetic diversity. It would be important to investigate the unique characteristics of this genotype such as their virulence in vitro and in vivo models and expand the screening efforts for more strains belonging to this lineage. Investigating pDST and transmission events using WGS, with higher depth coverage could further elucidate the role of the hSNPs in transmission inference.

The basis for TB genomic surveillance relies on the understanding of SNVs accumulation in *Mtb* and it offers more precise identification of recent transmission links compared to traditional contact investigations, providing more accurate data for public health responses. Moreover, on a broader scale, WGS allows the assessment of long-term transmission patterns in specific settings and the development of customized strategies for controlling transmission. The lack of a direct link between burden and transmission is underscored by the influence of previous outbreaks on the current incidence of TB. This emphasizes the importance of comprehensively understanding the dynamics of transmission and developing targeted TB control strategies [49]. It is possible that from 1998 to 2011 there was a delay in MDR-TB diagnostics as Xpert MTB/RIF was introduced in Brazil in 2014, enhancing the detection of TB and TB rifampicin-resistant (TB-RR) [50].

Despite an SNV distance of 28 between them, the two outlier isolates were positioned near the cluster, which exhibited an SNV distance of 12. This indicates that there are most likely several other cases not captured by this study. It is important to highlight that another WGS-based research in the same setting on Lineage 1 isolates among household contacts demonstrated that the SNP’s distance between patients living in the same house was more than nine SNPs (greater than five SNPs, as it is suggested for a cut-off of household contacts).

Studies have shown that conventional genotyping methods tend to overestimate transmission by clustering isolates that are linked through older rather than recent transmission chains (i.e.,: 30 years for 24-loci-MIRU-VNTR) [51].

Our study was constrained by its retrospective nature and a limited sample size. These limitations underscore the necessity of implementing genomic epidemiology surveillance in the Amazonian region of Brazil, which experiences a high incidence of TB, particularly because of the introduction of novel drugs such as bedaquiline, delamanid, linezolid, and pretomanid. WGS offers the highest level of resolution in identifying markers related to drug resistance and evolutionary changes. Moreover, it can be employed for surveillance to detect transmission and outbreaks [11,52].

Another limitation is the absence of more detailed clinical data such as disease progression and previous treatment history, which could have limited the algorithm’s ability to construct a more accurate transmission network. Default parameters from the TransFlow pipeline were also utilized (e.g., clock rate) and changes to these parameters could yield different transmission network structures. 

## 5. Conclusions

In this study, we performed a molecular characterization and drug resistance profiling of *Mtb* SIT2517/T1 isolates in Brazil. Phylogenetic analysis identified these isolates as sublineage 4.1.1.3, distinct from other representative *Mtb* Lineage 4 isolates from Brazil. The gDST and pDST agreed on the identification of the MDR profile for all isolates, including additional mutations being detected in the fabG1 gene for the G21387 isolate. Pyrazinamide resistance was detected in all isolates, and one isolate was classified as pre-XDR. This progression to pre-XDR could likely arise in cases that were missed and eventually progress towards XDR. Discrepancies between gDST and pDST were observed for pyrazinamide.

The geospatial distribution and transmission network analysis revealed a high likelihood of direct transmission in a relatively short space of time between confirmed contacts and the formation of two closely linked transmission chains within the cluster. G21384 and G21379 were the most likely index cases for these clusters with probable close and direct transmission between a person infected with the isolate G21379 and seven other patients.

In this investigation, one isolate (G21387), appears comparatively distant from the others and strongly suggests the existence of numerous overlooked cases and additional transmission chains and networks. The data support the necessity for investing in genomic surveillance in the Amazonian region of Brazil, particularly given its endemic status for TB.

## Figures and Tables

**Figure 1 microorganisms-12-01817-f001:**
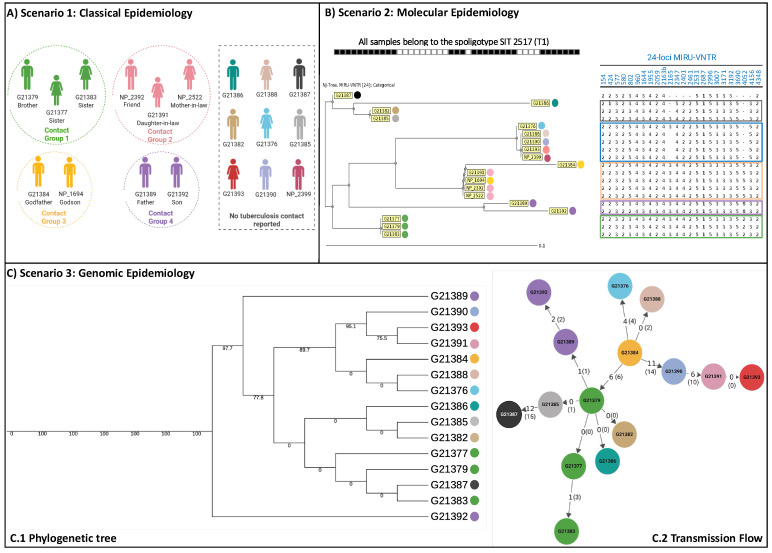
Analysis of 19 tuberculosis cases caused by *Mycobacterium tuberculosis* strains Shared International Type (SIT) 2517 sublineage T1 (SIT2517/T1) according to three scenarios. (**A**) Scenario 1, based on classical epidemiology, 10 cases are classified into four contact groups (sharing the same colour profile) and nine cases are not linked as contacts; (**B**) Scenario 2, based on molecular epidemiology, results from 24-loci MIRU-VNTR discriminated the strains with the same spoligotyping SIT2517/T1 results from 19 cases into five main groups; (**C**) Scenario 3, based on whole-genome sequencing (WGS) analysis it was possible to infer possible transmission links between 15 patients (two were excluded). One case (G21387) was not part of the transmission chain with more than 20 and 13 SNPs difference from strains G21388 and G21385, respectively.

**Figure 2 microorganisms-12-01817-f002:**
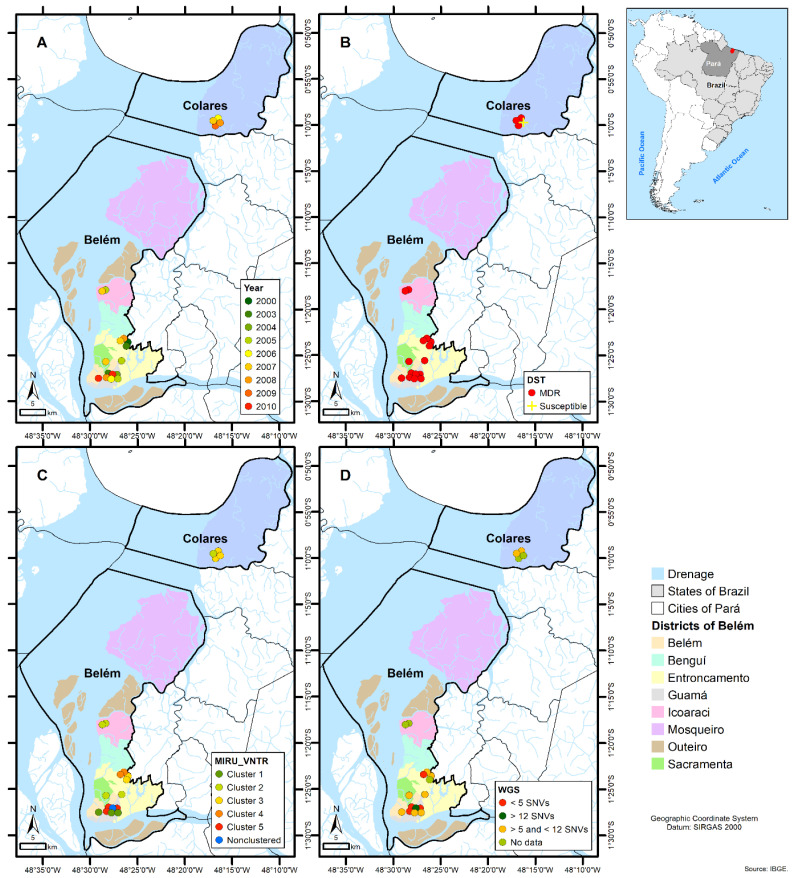
Geospatial distribution of the *Mycobacterium tuberculosis* strains Shared International Type (SIT) 2517 sub lineage T1 (SIT2517/T1) according to: (**A**) Year of bacterial isolation; (**B**) Phenotypic drug susceptibility testing (DST) profile; (**C**) 24-loci MIRU-VNTR clustering profile and (**D**) whole-genome sequencing clustering profile considering single nucleotide polymorphism (SNPs) distances.

**Table 1 microorganisms-12-01817-t001:** Genomic characteristics of *Mycobacterium tuberculosis* SIT 2517/T1 from State of Pará, Brazil versus drug susceptibility results and clustering analysis based on genotyping by 24-loci MIRU-VNTR and whole-genome sequencing.

Sample ID	Isolation Year	Mapped Reads ^1^ (%)	Number Mapped Reads ^1^	Median Coverage ^1^	pDST ^2^	gDST ^3^	MIRU-VNTR Cluster	WGS Cluster (SNPs) ^4^
G21377	2004	99.15	5,257,415	119	MDR	MDR	Cluster 5	<5
G21379	2003	99.17	5,924,978	135	MDR	MDR	Cluster 5	<5
G21383	2008	98.90	4,391,267	98	MDR	MDR	Cluster 5	<5
G21389	2007	99.16	7,025,682	158	MDR	MDR	Cluster 4	<5
G21376	2005	99.00	6,289,609	142	MDR	MDR	Cluster 2	>5 and <12
G21382	2010	99.14	6,303,757	144	MDR	MDR	Cluster 1	>5 and <12
G21384	2000	99.13	6,487,012	147	MDR	MDR	Cluster 3	>5 and <12
G21385	2006	99.09	6,198,316	140	MDR	MDR	Cluster 1	>5 and <12
G21386	2005	99.07	5,669,689	128	MDR	MDR	Cluster 1	>5 and <12
G21388	2007	99.11	6,528,434	148	MDR	MDR	Cluster 2	>5 and <12
G21391	2006	98.96	6,003,285	135	MDR	MDR	Cluster 3	>5 and <12
G21392	2008	98.02	6,177,468	119	MDR	MDR	Cluster 4	>5 and <12
G21393	2007	98.78	7,010,293	156	MDR	MDR	Cluster 2	>5 and <12
G21387	2010	99.03	7,018,299	158	MDR	MDR	No clustered	>12
G21390	2005	98.96	7,763,638	171	MDR	Pre-XDR	Cluster 2	>12
NP_1694	2003	No data	No data	No data	MDR	No data	Cluster 3	No data
NP_2392	2009	No data	No data	No data	MDR	No data	Cluster 3	No data
NP_2399	2007	No data	No data	No data	MDR	No data	Cluster 2	No data
NP_2522	2008	No data	No data	No data	S	No data	Cluster 3	No data

^1^ Against the *Mycobacterium tuberculosis* reference H37Rv (GenBank NC000962.3). ^2^ Phenotypic drug susceptibility test (pDST) based on proportion method. In this study, the results were: multidrug-resistant (MDR), when resistant at least to rifampicin and isoniazid, and drug-susceptible (S). ^3^ Genotypic drug susceptibility test (gDST) is based on Whole-genome sequencing (WGS) drug-resistance prediction. In this study, the results were: MDR and Pre-extensively drug-resistant (Pre-XDR), when resistant at least to rifampicin, isoniazid and any fluoroquinolone, a class of second-line anti-tuberculosis drug. ^4^ Whole-genome sequencing (WGS) cluster analysis based on single nucleotide polymorphism (SNPs).

## Data Availability

The raw sequencing data (fastq files) for this project is available on the National Center for Biotechnology Information (NCBI) repository under BioProject ID PRJNA609571.

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
