# Peer review of "A Genome-Focused Investigation Reveals the Emergence of a Mycobacterium tuberculosis Strain Related to Multidrug-Resistant Tuberculosis in the Amazon Region of Brazil"

_microorganisms, 2024, doi:10.3390/microorganisms12091817_

Round 1

Reviewer 1 Report

Comments and Suggestions for Authors

2 July, 2024.  Review of "A genome-focused investigation reveals the emergence of a Mycobacterium tuberculosis genotype related to multidrug-resistant tuberculosis in the Amazon region of Brazil"

This manuscript looks at a set of multi-drug resistant TB genomes from an outbreak in northern Brazil.  I cannot fully review this manuscript because only part of the data is available (15 of the 28 genomes sequenced have been deposited to GenBank).

The subject is interesting, but I found the reading difficult, in terms of clarity and jargon.  I understand that often specific technical terms are needed, but I found that even the abstract took a lot of work to read, even though I've been working with bacterial genomics for more than 20 years.  It took me awhile to figure out that 28 isolates were sequenced - this got lost with other details - perhaps the abstract could start with something along the lines that 28 genomes were sequenced - maybe in the 2nd sentence of the abstract?  And then go into more details later.

--> But I'm confused - what does it mean that of the 28 isolates, only 19 could be genotyped??  so nine of the genome sequences can't be genotyped?

According to Wikipedia, "The genotype of an organism is its complete set of genetic material."  So if one has the genome sequence, isn't that the genotype??  There needs to be some contextual explanation for what is meant here...

Further confusing is the following sentence that seems to imply that "MIRU-VNTR" is something different than genotyping - which I can kind of see - but the input to do this analysis is the genome sequence - but the Weniger et al. paper from 2010 (reference 29 in this manuscript) says in the abstract to their method that "Analysis and comparisons of genotypes can be based on MLVA-, spoligotype-, large sequence polymorphism and single nucleotide polymorphism data, or on a weighted combination of these markers."  so this IS using Genotype data...

MIRU-VNTR is never really defined in the current version of the manuscript - I did a search through the document, and can only see the acronym, but not what it stands for (see below for the definitions).  This can be easily fixed, and would be helpful for the reader to define this term.

MIRU: Mycobacterial Interspersed Repetitive Units

VNTR: Variable Number of Tandem Repeats

MLVA: Multi-Locus variable number tandem repeat analysis

I could only find 15 of the 28 genomes that were sequenced in NCBI (project ID: PRJNA609571) - where are the sequences for the remaining 13 genomes?

A few minor comments on the text:

Lines 65 through 68: This reads kind of awkward - perhaps break this into two sentences: 

"Tuberculosis (TB) is a preventable and curable disease, that remains....global epidemic_._"

and then continue:

"The first Mycobacterium tuberculosis (Mtb) _genome_ was published _more than a quarter of a century ago._ [4]"

Line 74: I am not sure what is being said here:

"The use of WGS in Mtb surveillance and research has demonstrated that the transmission of drug-resistant TB (DR-TB)... is one of the major driving forces of the DR-TB epidemic [9–11].

--> DR-TB is leading to the DR-TB epidemic??

Line 88: I think the word “is” needs removed

Lines 101 through 103: Sentence needs rewriting

Line 111: How is “large” defined? Is this truly a large study?

Lines 137 through 147: Are the settings for the software listed in a supplmental? Same as in line 372 - 376

Line 149: Mentioned repetitive regions? Would list specifically, if this is the case, PE/PEE regions of TB.

Table 1: SNPS greater than 5. More explanation is needed for this. In US, SNPs 5 or less is considered for clustering. Could be okay if Brazil is a high-burden type country. Don’t’ know. Would like to see why going to 12 SNPs is okay

See same in Lines 294, 328 - 332

Comments on the Quality of English Language

This paper needs to be improved in terms of English and also clarity of the scientific work.

Author Response

Reviewer 1 – Response (Manuscript ID: microorganisms-3093923)

Title “A genome-focused investigation reveals the emergence of a Mycobacterium tuberculosis genotype related to multidrug-resistant tuberculosis in the Amazon region of Brazil”.

Reviewer 1 comments

This manuscript looks at a set of multi-drug resistant TB genomes from an outbreak in northern Brazil.  I cannot fully review this manuscript because only part of the data is available (15 of the 28 genomes sequenced have been deposited to GenBank).

Response: To clarify, only a subset of the initial 15 isolates underwent WGS due to resource constraints. Of the original 28 isolates, nine were initially excluded from further analysis due to poor DNA quality or unsuccessful subculturing. Later, more 4 samples failed quality control for WGS. The remaining 15 isolates passed quality control and were subjected to WGS analysis. Please see the edited version: “Out of the total of 28 isolates of Mtb with genotype SIT2517/T1, 19 isolates (67.9%) were retrieved for further genotyping using a 24-loci MIRU-VNTR method. The remaining 9 strains could not be processed further. Issues included suboptimal DNA quality and limited cryopreserved stocks.  Out of 19 isolates, 10 possessed a complete set of 24 loci. The remaining isolates had varying degrees of incompleteness: six lacked two loci, two lacked one, and one isolate was missing a significant portion. The remaining nine isolates (32.2%) could not be genotyped due to either low DNA integrity (n=5; 17.9%) or the unavailability of DNA samples (n=4; 14.3%). We attempted subculture for these nine isolates but were unsuccessful. Among the 19 isolates, 17 underwent WGS, and two failed to pass quality control (coverage < 20X), so they were excluded from further WGS analysis. Therefore, a complete bioinformatic analysis was conducted on 15 isolates (54%) (Table 1).

The subject is interesting, but I found the reading difficult, in terms of clarity and jargon.  I understand that often specific technical terms are needed, but I found that even the abstract took a lot of work to read, even though I've been working with bacterial genomics for more than 20 years.  It took me awhile to figure out that 28 isolates were sequenced - this got lost with other details - perhaps the abstract could start with something along the lines that 28 genomes were sequenced - maybe in the 2nd sentence of the abstract?  And then go into more details later.

Response: We appreciate you bringing this to our attention. As you correctly noted, Microorganisms is a broad-scope journal that may have a general readership. To address this, we have carefully revised the manuscript to provide a more accessible introduction to the specialized terminology used in Mycobacterium tuberculosis (Mtb) genotyping. Specific terms and methods are now more explicitly defined and introduced within both the Abstract and Introduction sections.

According to Wikipedia, "The genotype of an organism is its complete set of genetic material."  So if one has the genome sequence, isn't that the genotype??  There needs to be some contextual explanation for what is meant here...

Response: In the field of Mycobacteriology, traditional genotyping methods such as spoligotyping, MIRU-VNTR, and RFLP analyze only specific genomic regions, distinct from whole-genome sequencing (WGS) which examines the entire genome. While classifications based on lineages, sub-lineages, and shared international types (SITs) are commonly referred to as “genotypes”. However, to prevent ambiguity, we have substituted “genotype” by “strain” throughout the title, abstract, and main text.

Further confusing is the following sentence that seems to imply that "MIRU-VNTR" is something different than genotyping - which I can kind of see - but the input to do this analysis is the genome sequence - but the Weniger et al. paper from 2010 (reference 29 in this manuscript) says in the abstract to their method that "Analysis and comparisons of genotypes can be based on MLVA-, spoligotype-, large sequence polymorphism and single nucleotide polymorphism data, or on a weighted combination of these markers."  so this IS using Genotype data... MIRU-VNTR is never really defined in the current version of the manuscript - I did a search through the document, and can only see the acronym, but not what it stands for (see below for the definitions).  This can be easily fixed, and would be helpful for the reader to define this term.

Response: We have refined the abstract and introduction to provide a clearer explanation of MIRU-VNTR (Mycobacterial Interspersed Repetitive Units Variable Number of Tandem Repeats). While MIRU-VNTR is a specific type of MLVA (Multi-Locus Variable Number Tandem Repeat Analysis), it is commonly referred to simply as MIRU-VNTR.

I could only find 15 of the 28 genomes that were sequenced in NCBI (project ID: PRJNA609571) - where are the sequences for the remaining 13 genomes?

Response: Only 15 of the 28 isolates were subjected to WGS. We amended the manuscript to make this more clear. See first paragraph of the Results section.

A few minor comments on the text:

Lines 65 through 68: This reads kind of awkward - perhaps break this into two sentences:  "Tuberculosis (TB) is a preventable and curable disease, that remains....global epidemic_._" and then continue: "The first Mycobacterium tuberculosis (Mtb) _genome_ was published _more than a quarter of a century ago._ [4]"

Response: This entire section of the introduction was rewritten and rephrased for clarity, as per suggestion.

 Line 74: I am not sure what is being said here: "The use of WGS in Mtb surveillance and research has demonstrated that the transmission of drug-resistant TB (DR-TB)... is one of the major driving forces of the DR-TB epidemic [9–11]. --> DR-TB is leading to the DR-TB epidemic??

Response: Thank you for this observation! The emphasis here is transmission of DR-TB, rather the acquisition. We removed the second instance of DR in the text to focus on the dynamic of TB epidemic being shaped by the transmission of DR strains (lines 113-116).

Line 88: I think the word “is” needs removed.

Response: corrected.

Lines 101 through 103: Sentence needs rewriting.

Response: sentence edited.

Line 111: How is “large” defined? Is this truly a large study?

Response: In the context of Mtb surveillance in Brazil, we consider this large. More than a 980 strains were assessed. However, to avoid subjective, we removed the word “large”.

Lines 137 through 147: Are the settings for the software listed in a supplmental? Same as in line 372 - 376

Response: We appreciate you highlighting the importance of reproducibility. As you suggested, we have clarified in both sections that 'Default parameters were utilized for all software packages' to enhance transparency and replicability.

Line 149: Mentioned repetitive regions? Would list specifically, if this is the case, PE/PEE regions of TB.

Response: Amended in the text.

Table 1: SNPS greater than 5. More explanation is needed for this. In US, SNPs 5 or less is considered for clustering. Could be okay if Brazil is a high-burden type country. Don’t’ know. Would like to see why going to 12 SNPs is okay. See same in Lines 294, 328 – 332

Response: We acknowledge this comment and agree that SNP thresholds are arbitrary. Below 12 SNPs generally refers to a cluster where the isolates are part of a transmission network. Recently, the importance of factoring in variants associated with drug-resistance has gained traction (doi: 10.1099/mgen.0.000815.). From our analysis, all of our closely related isolates had the same identical DR-associated variants. It is highly unlikely that these were all due to homoplastic events. We have amended sections in the Discussion to highlight this point.

Comments on the Quality of English Language: This paper needs to be improved in terms of English and also the clarity of the scientific work.

Response: We have rewritten a large section and improved the grammar.

Reviewer 2 Report

Comments and Suggestions for Authors

The study was performed with 28 M.tb strains collected  more than 10 years ago – in 2008–2011.

The text requires a more clear description of how the dataset was created. It’s not clear if the authors analyzed all strains that belonged to SIT3517 in the country or only a portion?

What happens with this genotype in Brazil nowadays? How often is it identified?

Why was only part of the SIT3517 strains genotyped with MIRU-VNTR?

How did you decide which samples had to be sequenced using the NGS approach?

It is mentioned that the average sequencing depth was 20x, but further, you describe the filtration of SNVs and mention that SNVs with a minimum 30x coverage were included in the study. I see a contradiction here, because it is obvious that a lit of SNVs are removed from the analysis. This could affect the quality of the result.

The MIRU-VNTR method is not described.

Why were only 15 isolates subject to complete bioinformatic analysis?

Why 4 samples mentioned in the table 1 don’t have dDST data?

Why didn’t you sequence the only drug-susceptible strain (2522)? Is it possible to perform WGS with this sample?

Figure 1: The meaning of the drawing is not clear.

  1. A) Why is there so little data on the epidemiological scenario? This does not reflect the transmission path of the 28 samples. 
  2. B) The signatures of the samples are not readable.
  3. C) There are no bootstrap values, and the sample numbers on the transmission tree are not visible.

Figure 2.D. It is not clear what each spot means. Is it one sample or a group of samples? If it is a sample, what does the distance in SNPs measure?

Author Response

Reviewer 2 – Authors Reply (Manuscript ID: microorganisms-3093923)

Title “A genome-focused investigation reveals the emergence of a Mycobacterium tuberculosis genotype related to multidrug-resistant tuberculosis in the Amazon region of Brazil”.

Reviewer 2 comments

This study was performed with 28 M.tb strains collected  more than 10 years ago – in 2008–2011. The text requires a more clear description of how the dataset was created. It’s not clear if the authors analyzed all strains that belonged to SIT3517 in the country or only a portion?

Response: So far only he state of Pará has reported this genotype SIT2517 in Brazil and World. We have improved the Study Population description for more clarity: “Between 1998 and 2010, we conducted a population-based study on 980 Mtb isolates isolated obtained from pulmonary TB patients in Pará, Brazil. All isolates underwent spoligotyping and 28 isolates exhibiting SIT 2517/T1 profile were selected for this study [15] for further investigation based on MIRU-VNTR and WGS.”

What happens with this genotype in Brazil nowadays? How often is it identified?

Response: So far, it has not been detected and described again, since WGS is not performed routinely here. As part of a follow up project, currently we will start performing WGS within the context of a TB genomic surveillance network and we will observe if this genotype is circulating in Brazil, its drug resistance profile and transmission.

Why was only part of the SIT3517 strains genotyped with MIRU-VNTR?

Response: Because some samples were lost due to poor DNA quality and not being viable when subcultured. We have amended the first paragraph of the Results section for clarity: “Out of the total of 28 isolates of Mtb with genotype SIT2517/T1, 19 isolates (67.9%) were retrieved for further genotyping using a 24-loci MIRU-VNTR method. The remaining 9 strains could not be processed further. Issues included suboptimal DNA quality and limited cryopreserved stocks.  Out of 19 isolates, 10 possessed a complete set of 24 loci. The remaining isolates had varying degrees of incompleteness: six lacked two loci, two lacked one, and one isolate was missing a significant portion. The remaining nine isolates (32.2%) could not be genotyped due to either low DNA integrity (n=5; 17.9%) or the unavailability of DNA samples (n=4; 14.3%). We attempted subculture for these nine isolates but were unsuccessful. Among the 19 isolates, 17 underwent WGS, and two failed to pass quality control (coverage < 20X), so they were excluded from further WGS analysis. Therefore, a complete bioinformatic analysis was conducted on 15 isolates (54%) (Table 1).”

How did you decide which samples had to be sequenced using the NGS approach?

Response: Ideally all samples should be WGS sequenced, but those with poor DNA quantity based on qubit were excluded. As MIRU-VNTR requires more DNA input, we lost samples after MIRU-VNTR method being performed.

It is mentioned that the average sequencing depth was 20x, but further, you describe the filtration of SNVs and mention that SNVs with a minimum 30x coverage were included in the study. I see a contradiction here, because it is obvious that a lit of SNVs are removed from the analysis. This could affect the quality of the result.

Response: This is an oversight from our side. We have corrected this in the manuscript. The 30X coverage refers to a minimum of 30X depth of coverage of a variant at a specific genomic position for it to be accepted as a variant. We removed the 20X sequencing depth line as this is irrelevant in any case since the depth of coverage relative to the reference genome is more important. As is seen from the table. We have amended the text for clarity on this regarding.

The MIRU-VNTR method is not described.

Response: We’ve included a definition and description of this genotyping method in the introduction and methods sections. The method is described within the subtopic “Culture, Drug Susceptibility and Molecular Tests”.

Why were only 15 isolates subject to complete bioinformatic analysis?

Response: All 17 isolates that were selected for WGS were bioinformatically analysed, but two completely failed due to almost no coverage.

Why 4 samples mentioned in the table 1 don’t have dDST data?

Response: The 4 samples without gDST is because WGS was not performed for these and therefore no variants associated with drug resistance could be identified.

Why didn’t you sequence the only drug-susceptible strain (2522)? Is it possible to perform WGS with this sample?

Response: This would be ideal, but these isolates have lost viability and can not be sequenced anymore. Based on the scenarios we indicated and the fact that this strain appears to transmit both within households as well as in the community, this likely could have been a more distant, less virulent strain. Or, gDST might have revealed inaccurate pDST predictions.

Figure 1: The meaning of the drawing is not clear.

  1. A) Why is there so little data on the epidemiological scenario? This does not reflect the transmission path of the 28 samples. 
  2. B) The signatures of the samples are not readable.
  3. C) There are no bootstrap values, and the sample numbers on the transmission tree are not visible.

Response: We have added in the bootstraps. For most branches, the isolates are too closely related and therefore could have been placed on any of the branches in most of the clade. This highlights how closely related these strains are.

Figure 2.D. It is not clear what each spot means. Is it one sample or a group of samples? If it is a sample, what does the distance in SNPs measure?

Response: Each circle represents an isolate. And the SNP distance between the closest isolates. The arrows indicated predicted direction of transmission, as calculated by TransFlow.

Round 2

Reviewer 1 Report

Comments and Suggestions for Authors

I think this version of the manuscript is substantially improved. 

Author Response

Dear Reviewer,

We would like to express our sincere gratitude for your time and valuable feedback. Your insightful comments have significantly enhanced the quality of our research.